# Genome-wide Estrogen Receptor-α activation is sustained, not cyclical

Andrew N Holding*, Amy E Cullen, Florian Markowetz

Cancer Research UK Cambridge Institute, University of Cambridge, Cambridge, United Kingdom

**Abstract** Estrogen Receptor-alpha (ER) drives 75% of breast cancers. Stimulation of the ER by estra-2-diol forms a transcriptionally-active chromatin-bound complex. Previous studies reported that ER binding follows a cyclical pattern. However, most studies have been limited to individual ER target genes and without replicates. Thus, the robustness and generality of ER cycling are not well understood. We present a comprehensive genome-wide analysis of the ER after activation, based on 6 replicates at 10 time-points, using our method for precise quantification of binding, Parallel-Factor ChIP-seq. In contrast to previous studies, we identified a sustained increase in affinity, alongside a class of estra-2-diol independent binding sites. Our results are corroborated by quantitative re-analysis of multiple independent studies. Our new model reconciles the conflicting studies into the ER at the TFF1 promoter and provides a detailed understanding in the context of the ER's role as both the driver and therapeutic target of breast cancer.
DOI: https://doi.org/10.7554/eLife.40854.001

## Introduction

The study of the Estrogen Receptor-α (ER) has played a fundamental role in both our understanding of transcription factors and cancer biology. The ER is one of a family of transcription factors called nuclear receptors. Nuclear receptors are intra-cellular and, on activation by their ligand, typically undergo dimerisation and bind to specific DNA motif (for ER: Estrogen Response Elements; EREs). On the chromatin, the nuclear receptor recruits a series of cofactors and promotes the basal transcription mechanism at either nearby promoters or through chromatin loops from distal enhancers. Because of the minimal nature of these systems relative to other signaling pathways, nuclear receptors have become a model system for transcription factor analysis. Simultaneously, the role of nuclear receptors as drivers in a range of hormone dependent cancers has led to focused studies in the context of the disease.

Previously, it was reported that the ER and key cofactors followed a cyclical pattern in breast cancer cell lines with maximal binding at 45 min after stimulation with estra-2-diol (*Shang et al., 2000*; *Métivier et al., 2003*). Similar results were also reported for the AR after activation with DHT (*Kang et al., 2002*) and several follow-up studies exist looking at single genomic *loci* (*Herynk et al., 2010*; *Luo et al., 2005*; *Shao et al., 2004*; *Burakov et al., 2002*). However, subsequent genome-wide studies have provided little further detail on the specific nature of the proposed kinetics of ER binding being either limited in the number of replicates or lacking temporal resolution (*Honkela et al., 2015*; *wa Maina et al., 2014*; *Dzida et al., 2017*; *Guertin et al., 2014*). In our own network analysis (*Holding et al., 2018*), we focused on 0, 45 and 90 min and found no significant reduction in ER signal at 90 min. In the same study, quantitative proteomic analysis of ER interactions at the same time intervals by qPLEX-RIME (*Papachristou et al., 2018*) shows no significant difference in terms of ER interactions at 45 and 90 min. These conflicting results have so far not been resolved.

*For correspondence:
andrew.holding@cruk.cam.ac.uk

**Competing interests:** The authors declare that no competing interests exist.

**eLife digest** Breast cancer is the most common type of cancer worldwide. The hormone estrogen drives the growth of 70% of breast cancer tumors. This form of breast cancer is called estrogen receptor positive (ER+) breast cancer. In the early 2000s, several scientists found that some genes in ER+ breast cancers turn on and off in 90-minute cycles. Moreover, when the estrogen receptor binds to the DNA in the nucleus of a cell, it activates nearby genes causing the tumor cells to grow and divide.

Learning more about how cancer cells respond to estrogen is very important. Many cancer drugs block estrogen to stop its tumor growth promoting effects. But the initial studies of estrogens effects were only able to look at how estrogen affected a small number of genes. Newer genome sequencing technologies allow scientists to study the effects of estrogen on more genes and provide more detailed information.

Using these cutting-edge technologies, Holding et al. show that the 90-minute cycles found in the previous studies are likely artefacts of older techniques and lacking controls. The new experiments used a newer technique called parallel factor ChIP-seq to look at how all genes respond to the estrogen receptor. Then, Holding et al. reanalyzed data published in the previous studies and found that they were often contradictory and inconsistent.

None of the genes – not even the ones looked at in earlier studies – were expressed in 90-minute cycles like the previous studies suggested. Instead, the expression of the genes was variable, which may make the cell even more responsive to estrogen. The previous reports of the 90-minute cycles are most likely explained by a bias of the human eye of finding patterns in a highly variable process that do not hold up to statistical analysis.

Better understanding how estrogen influences genes and cell growth is essential to developing better treatments for ER+ breast cancer. This includes ruling out ideas that may be incorrect or misleading. These findings help resolve why not all studies have found estrogen receptor driven cycles of gene expression, and will provide researchers with a better foundation for future studies.
DOI: https://doi.org/10.7554/eLife.40854.002

Routinely used assays to measure protein binding to chromatin are based on Chromatin Immuno-precipitation (ChIP). A major challenge to monitoring ER activation through ChIP is the normalization of the ChIP signal — either genome-wide with next generation sequencing or at individual loci by qPCR — as the standard protocols do not control for a significant number of confounding factors including the efficiency of the immunoprecipitation step. In the case the of the two original studies (*Shang et al., 2000*; *Métivier et al., 2003*), the data only provided limited controls in this regard. An alternative method that has been applied to normalize ChIP-seq data is to use the maximal read count obtained at each individual site across each time point (*Guertin et al., 2014*); however, this method is at the expense of monitoring the magnitude of ER binding and gives equal weight to low read count peaks and more robust data from stronger binding sites.

In the context of these challenges, we applied two strategies to robustly and accurately monitor the process of nuclear receptor binding to chromatin on activation. The first strategy was to increase the number of replicates. We generated sample data for six independent isogenic experiments to enable better characterization of the variance within the data. This strategy provided an unprecedented level of information regarding ER activation with twice the level of replication used in previous ChIP-qPCR studies (*Métivier et al., 2003*) and a significant improvement on previous single replicate genome-wide studies. The second strategy was to use our recently developed method for precise quantification of binding, Parallel-Factor ChIP (pfChIP) (*Guertin et al., 2018*), which uses an internal control for quantitative differential ChIP-seq. Combined, these two strategies enabled us to undertake the most comprehensive and precise analysis of ER activation to date.

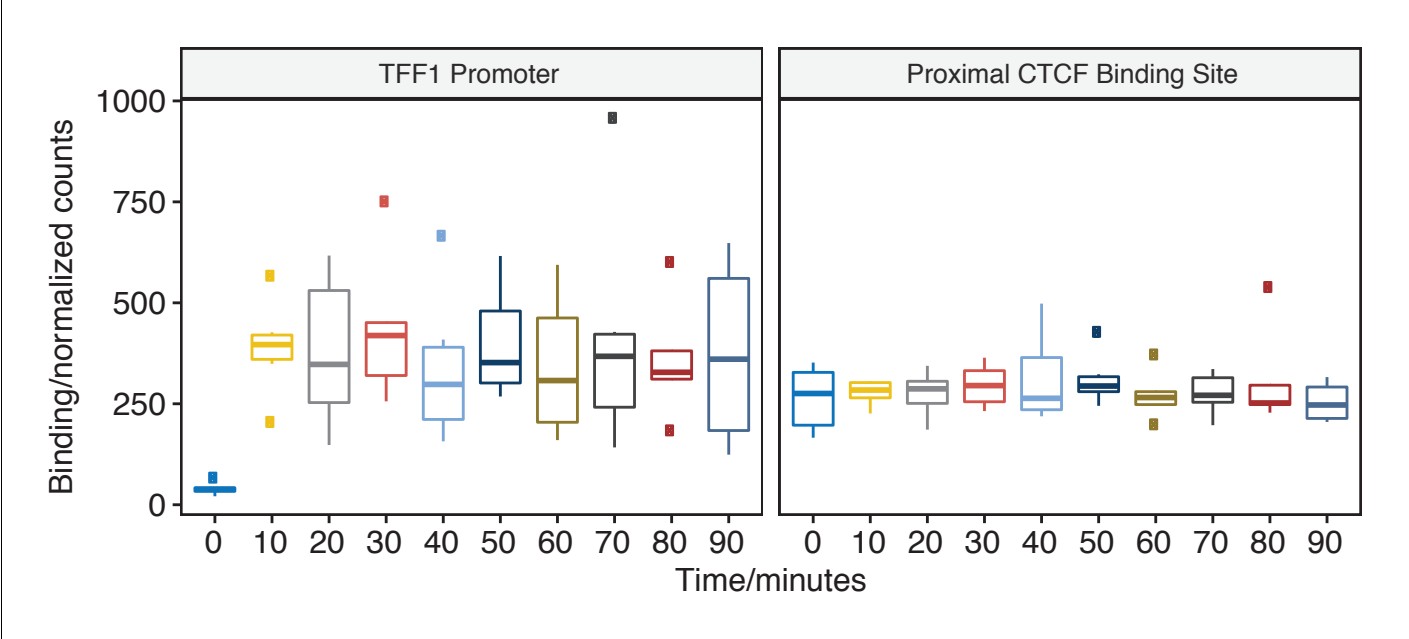

**Figure 1.** pfChIP-seq signal at the TFF1 promoter and proximal CTCF binding site. Binding of ER at the TFF1 promoter has been the classical focus of study before genome-wide technology and the predicted site for oscillations in ER binding. ER binding is minimal at 0 min; however, by 10 min, the ER has rapidly and robustly bound to give a sustained signal at the TFF1 promoter. In contrast, the closest CTCF binding site demonstrates a constant, estra-2-diol-independent, signal with significantly less variance. Pairwise comparison found no significant changes in binding at the TFF1 promoter (t-test, two-sided, FDR < 0.05) except for when comparing against the 0 min time point.

DOI: https://doi.org/10.7554/eLife.40854.003

The following figure supplements are available for figure 1:

**Figure supplement 1.** TFF1 Promoter and Enhancer.

DOI: https://doi.org/10.7554/eLife.40854.004

**Figure supplement 2.** Normalization plots for each time point as generated by Brundle (*Guertin et al., 2018*).

DOI: https://doi.org/10.7554/eLife.40854.005

**Figure supplement 3.** Line plot of pfChIP-seq signal at the TFF1 promoter and proximal CTCF binding site.

DOI: https://doi.org/10.7554/eLife.40854.006

**Figure supplement 4.** Plot of pfChIP-seq signal at the RARA promoter and proximal CTCF binding site.

DOI: https://doi.org/10.7554/eLife.40854.007

## Results

### Measurement of genome Copy-Number discordance

We measured ER-binding in MCF7 cells, a widely used model system for ER biology. To maximize the reproducibility of our results, MCF7 cells were grown from ATCC stocks, keeping passaging to a minimum, and the cell line origin was confirmed by STR genotyping. Additionally, to ensure the MCF7 cell line did not show significant genetic drift during culturing within our laboratory, we applied CellStrainer (*Ben-David et al., 2018*) to the input data from our ChIP-seq experiments. The fraction of genome with copy-number discordance was estimated at 0.2787, within the range of 0 to 0.3 as published by CellStrainer's developers to ensure similar therapeutic response.

### Visualization of raw data

Sequencing reads from the analysis of 60 pfChip-seq samples targeting ER and six input samples were demultiplexed and aligned to the Homo sapiens GRCh38 reference assembly. Visual inspection of the data using the Integrative Genomics Viewer (IGV) Viewer (*Robinson et al., 2011*) confirmed enrichment at known ER binding sites (exemplified by TFF1 in *Figure 1—figure supplement 1*) and the presence of previously reported CTCF control peaks (*Guertin et al., 2018*). From visual

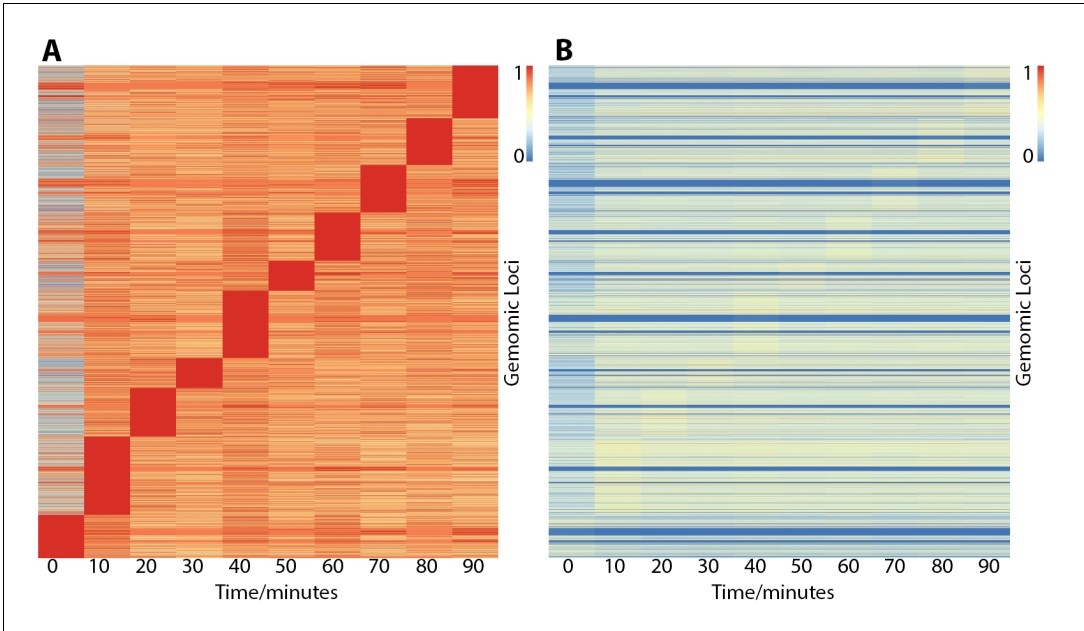

**Figure 2.** Heatmaps showing ER binding affinity from 0 to 90 min after stimulation with estra-2-diol normalized in two different ways. Row order is the same in both plots. (**A**) Normalized by row to time point with maximal binding. Data suggests that genomic loci may influence the time point maximal binding; however, normalizing to CTCF control peaks (**B**) demonstrates the effect is potentially overemphasized by normalization choice and that binding affinity is the biggest variable. In contrast, both plots (**A and B**) show minimal ER binding affinity is found at 0 min, consistent with the literature response of MCF7 cells to treatment of estra-2-diol.
DOI: https://doi.org/10.7554/eLife.40854.008

The following figure supplements are available for figure 2:

**Figure supplement 1.** (Left) Analysis of each time point to the previous time point only shows a significant numbers of changes in ER binding between 0 and 10 min.
DOI: https://doi.org/10.7554/eLife.40854.009

**Figure supplement 2.** (Left) Class average of ER binding in the block of binding events that show maximum binding at 0 min in *Figure 2*.
DOI: https://doi.org/10.7554/eLife.40854.010

inspection, pfChIP-seq samples qualitatively showed minimal ER binding at 0 min while CTCF binding was constant at all time points.

## Parallel-Factor normalization

Peak count data from CTCF binding sites were used to normalize between conditions as these sites have previously been shown to be unchanged in response to estrogen (*Ross-Innes et al., 2011*), with >70 000 binding sites discovered across all samples and >50 000 CTCF binding sites found in over 50% of samples. Analysis after normalization of the raw data showed similar levels of variability in terms of signal (*Figure 1—figure supplement 2*) as we saw when developing the pfChIP method (*Guertin et al., 2018*). The resultant normalized binding matrix of ER binding was used for all downstream analyses and is provided as *Supplementary file 1*.

## ER binding at the TFF1 promoter

Normalized count data for the TFF1 promoter showed that on activation with estra-2-diol the ER rapidly (in less than 10 min) binds the TFF1 promoter. Binding after this time point shows no significant changes (*Figure 1*). Analysis of the data by individual replicates (*Figure 1—figure supplement 3*) did not demonstrate evidence of oscillatory binding in individual replicates either with a period of 90 min period or an alternative frequency.

Comparison of the variance in the ER binding after induction shows that there is significantly more variance (F-test, time points >= 10 min, p-value $< 1 \times 10^{-10}$) in the ER binding data than in

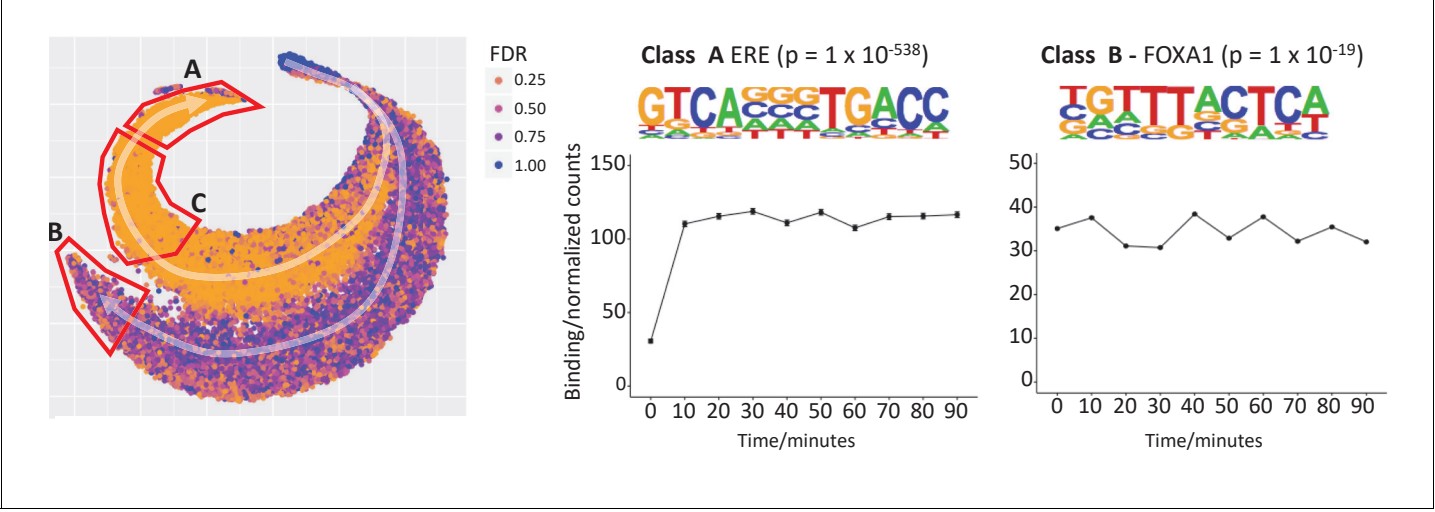

**Figure 3.** t-SNE plot to explore temporal patterns in ER binding affinity. Two trajectories, A and B, are highlighted with white arrows and starting a single cluster of peaks. Points are colored by FDR value computed by Brundle/DiffBind for the 0 vs 10 min contrast. Trajectory A demonstrates increasing ER affinity in response to estra-2-diol at 10 min. Trajectory B shows increasing affinity for all times points, that is estra-2-diol independent binding, but the maximum signal is of a lower intensity than that of Trajectory A. De novo motif analysis for Class A (the peaks found at the end of trajectory A) gave strongest enrichment for the ERE ($p = 1 \times 10^{-538}$). The same analysis of Class C provided a partial ERE (not shown), consistent with ER affinity being a function of how conserved the ER binding site is with respect to ideal ERE. Analysis of Class B gave FOXA1 as the most significantly enriched motif ($p = 1 \times 10^{-19}$).

DOI: https://doi.org/10.7554/eLife.40854.011

The following figure supplements are available for figure 3:

**Figure supplement 1.** Multiple t-SNE plots of ER binding affinity changes in response to estra-2-diol at increasing perplexity (top left of each figure).
DOI: https://doi.org/10.7554/eLife.40854.012
**Figure supplement 2.** Analysis of the ER data stream gave similar profiles at the TFF1.
DOI: https://doi.org/10.7554/eLife.40854.013

CTCF binding between replicates. In contrast, pairwise F-test (two-sided, FDR < 0.05) for ER binding at all time points showed no significant difference in the variance for any comparison. As the variance of CTCF binding in pfChIP-seq is a good estimator of the technical variance, the most likely source of increased variance in ER binding is therefore biological. These findings were validated through analysis of the RARA promoter and proximal CTCF peaks (*Figure 1—figure supplement 4*), which gave consistent results to those seen at the TFF1 promoter.

## Locus specific variation in maximal ER binding affinity

Previously, ER binding sites were shown to reach maximum occupancy at different time points depending on genomic location, revealing a P300 squelching mechanism at early time points (*Guertin et al., 2014*). Therefore, to provide a partial validation of this study, we applied the same principles of their analysis to our data, that is normalizing in the time-space setting maximum occupancy to 1. Consistent with the previous study, the two time points with the largest numbers of sites reaching maximal occupancy in both data sets were at 10 and 40 min (*Figure 2A*). As the remaining time points were unique to the individual data sets, these could not be directly compared.

While grouping by maximum occupancy in *Figure 2A* was essential to highlight these features in the context of Guertin *et al.*'s previous study, in our analysis we found this method distorts the data and the effects that drove the appearance of blocks are not statistically significant in our dataset (*Figure 2—figure supplement 1*) with the exception of the 0 to 10 min contrast.

As far as we are aware, the loss on ER binding on activation with estra-2-diol is unprecedented, and therefore the presence of a block of ER sites with maximal binding at 0 min warranted deeper investigation. Analysis of the class average (*Figure 2—figure supplement 2A*) showed that the variance of this class is much greater than the decrease seen between 0 and 10 min. A more detailed analysis of the individual trajectories of each binding site (*Figure 2—figure supplement 2B*) showed

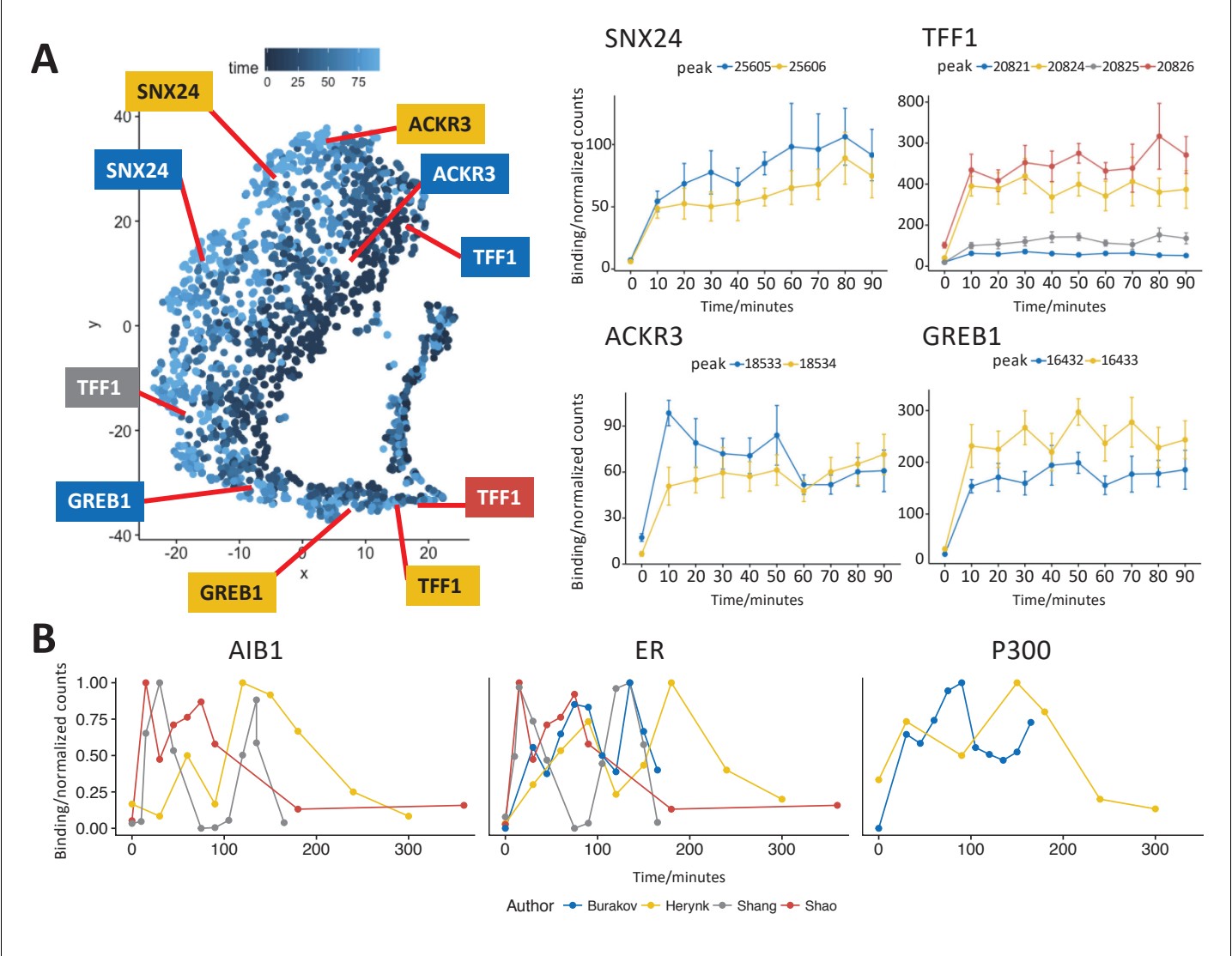

**Figure 4.** t-SNE analysis of the most strongly bound ER sites (Class A) and comparison to previously published data. (A) t-SNE plot of Class A from *Figure 3*, points colored by time of maximum value. Profiles for binding sites near the transcription start site of two well studied ER target genes, *TFF1* and *GREB1*, gave a robust sustained response to estra-2-diol. Binding sites near *SNX24* and *ACKR3* TSS are shown to examples of ER binding affinity profiles that indicate potential early or late maximal binding. Peak coordinates are provided in *Supplementary file 1*. (B) Analysis of ER and cofactor binding at the TFF1 promoter in four studies (*Burakov et al., 2002*; *Herynk et al., 2010*; *Shang et al., 2000*; *Shao et al., 2004*). Data was either read directly from plots within the original publication or using ImageJ (*Schneider et al., 2012*) to calculate band density. To ensure data was comparable, data was normalized to the maximum value and all studies were chosen for replicating the conditions in Shang *et al.*'s original study.
DOI: https://doi.org/10.7554/eLife.40854.014

a similar result, with the maximal binding at time zero appearing marginal and in all cases. We therefore concluded that maintaining the group of the blocks between *Figure 2A and B* visually overemphasized this feature within the data.

pfChIP-seq allowed us to improve on the previous study by directly normalizing the data to the internal control. The resultant binding matrix provided quantification of the absolute binding affinity at each time point (*Figure 2B*).

Comparison of *Figure 2A and B* demonstrates the effects of different data normalization strategies. The relative normalization to maximum binding emphasizes binding maxima (red blocks in *Figure 2A*) while the absolute normalization to an internal control shows that these maxima are very shallow, barely visible in *Figure 2B*, and other features dominate the data. A few genes show very

high levels of ER binding (visible as thin red lines in *Figure 2B*), while most genes show intermediate levels and some very low levels (blue lines). These different levels of ER binding are preserved over time, with only time point 0 showing very low levels for all genes.

## Visualizing temporal ER binding affinity

To elucidate potential different temporal responses to ER activation by estra-2-diol, we applied t-SNE (*Maaten and Hinton, 2008*), a widely used method for dimensionality reduction and data visualization (*Figure 3*). Each dot in the plot represents a binding site over time, that is one row in the binding matrix shown in *Figure 2B*. We colored each dot by the false discovery rate (FDR; (*Benjamini and Hochberg, 1995*)) for the change in ER affinity between 0 to 10 min. This analysis revealed two major trajectories of binding sites in the data, one dominated by low FDR (orange) and one by high FDR (blue). Both trajectories saw an increasing affinity in the direction of the white arrow. This pattern was stable for a wide range of perplexity, the main t-SNE parameter (*Figure 3—figure supplement 1*).

We named the estra-2-diol responsive trajectory A, and the estra-2-diol independent trajectory B. The set of genomic sites found at the end of each trajectory were named Class A and B respectively. Motif analysis of Class A peaks demonstrated significant enrichment for the full estrogen response element (ERE, (*Klein-Hitpass et al., 1986*)), while Class B gave enrichment for the *FOXA1* binding site. Analysis of Class C (i.e. weaker responding genes on trajectory A) gave a partial ERE match, suggesting a greater divergence from the ERE motif and consistent with the lower levels of ER affinity found on ER activation at these sites (*Driscoll et al., 1998*).

Average binding profiles were computed for both Class A and Class B. Class A showed minimal binding at 0 min followed by a robust response before 10 min, the binding affinity then remained similar for the remaining time points. In contrast, Class B displayed estra-2-diol independent binding at 0 min and average ER binding affinity saw no significant changes between time points. Class C gave a similar profile to Class A (not shown), but with reduced amplitude. The average amplitude of the binding from 10 to 90 min displayed a greater ER affinity for Class A then Class B.

Genomic regions enrichment of annotations tool (GREAT) analysis (*Welch et al., 2014*) of Class B binding sites (*Supplementary file 2*) identified the enrichment of six amplicons previously identified from the analysis of 191 breast tumor samples, $q = 5.6 \times 10^{-41}$ to $q = 3.3 \times 10^{-8}$, (*Nikolsky et al., 2008*) and a set of genes upregulated in luminal-like breast cancer cell lines compared to the mesenchymal-like cell lines, $q = 1.9 \times 10^{-13}$, [*Charafe-Jauffret et al., 2006*]).

Undertaking the same analysis of the ER only ChIP-seq data stream gave very similar results to that of pfChIP-seq analysis, confirming that any potential cycling is not suppressed by the method (*Figure 3—figure supplement 2*). As with the pfChIP-seq analysis, no clear cycling was seen for the individual replicates (*Figure 3—figure supplement 2C*).

## Analysis of class A ER binding sites

Class A binding sites showed the strongest response to estra-2-diol, the greatest enrichment of the estrogen response element and contained the classical ER binding site at *TFF1*. We therefore focused further analysis on these peaks to minimize confounding factors. A t-SNE plot of only Class A sites (*Figure 4A*) did not provide distinct clustering of points. Partial separation was seen on the basis of time point of maximal binding (left to right) and amplitude (approximately top to bottom).

As the class profiles may average out site-specific oscillatory kinetics, we undertook analysis of individual ER binding sites. Peaks were annotated on the basis of the nearest Transcription Start Sites (TSS) and profiles for key ER target genes *TFF1* and *GREB1* were generated. As previously seen in *Figure 1—figure supplement 1*, ER binding at *TFF1* was stable after induction. The same response was seen at the *TFF1* enhancer (dark red). Analysis of ER binding proximal to *GREB1* again showed a robust and unidirectional response to estra-2-diol.

Profiles of ER binding that showed either early or late maximal ER affinity were individually investigated. Binding near the TSS of *SNX24* and *ACKR3* are provided as representative examples.

## Quantitative re-analysis of independent studies

Given we found a robust and stable response to ER activation by estra-2-diol in contrast to the cyclical response previously described (*Shang et al., 2000*), we reviewed studies that have investigated

ER binding at the *TFF1* promoter. Several studies either used a different promoter (*Park et al., 2005*), factor (*Li et al., 2003*) or estra-2-diol concentration/include α-amanitin (*Métivier et al., 2003*).

By manually reviewing the first 1000 citations of (*Shang et al., 2000*), we identified several studies (*Burakov et al., 2002*; *Herynk et al., 2010*; *Shao et al., 2004*) that undertook the same analysis in the MCF7 cell line, with the same concentration of estra-2-diol, same crosslinking time scale, and at the same promoter as the best datasets for comparison with each other and to our dataset.

Since the numerical values of ER binding occupancy were not available for these studies, we read the values off the provided charts or undertook image analysis of figures (*Supplementary file 4*).

Comparison of the data from all four studies gave little or no consistency in the temporal profile of ER, AIB1 and P300 binding at these sites (*Figure 4B*). Quantitative image analysis presents limitations in reestablishing the exact values without the primary data; however, the primary data was not available. In lieu of this, analysis of the published data was considered adequate as the relative intensities will be preserved. Interpretation is further hindered as these studies only report a single replicate for analysis, thereby making it impossible to quantify uncertainty in the data. Therefore, there is no consistent evidence for cycling in the studies using the same conditions as the original observation.

## Discussion

By undertaking six biological replicates and incorporating an internal control with pfChIP, we have produced the most comprehensive analysis to date of ER binding over the first 90 min after stimulation with estra-2-diol. We found the sites at which we detected ER binding on the chromatin follows two distinct trajectories, either the rapid activation within 10 min followed by a stable response or ligand independent binding.

Enrichment of the FOXA1 motif in the strongest ligand-independent/Class B sites supports our hypothesis: that these are as a result of ER interactions at these sites. Importantly, the de novo motif analysis did not find the presence of the CTCF motif, confirming that they are not an artifact of utilizing CTCF to normalize *via* the pfChIP-seq method. Analysis of the Class B binding sites with GREAT (*Welch et al., 2014*), *Supplementary file 2*, gave enrichment for 6 out of 30 ER regulated amplicons identified in a previous study of 191 breast cancer tumor samples (*Nikolsky et al., 2008*). On the basis that no ERE was found at Class B sites and that the affinity of ER at these sites was less than at estra-2-diol response sites, we propose that these sites represent open regions of chromatin where ER can be recruited by other transcription factors in the absence of its own ligand. However, these interactions are weak, and very likely transient, as the average binding affinity for Class B sites is similar in level (a normalized read count of 30–40) to the binding before activation at Class A binding sites, but greater than Class C binding sites ($\approx$ 10 normalized reads increasing to $\approx$ 40 on activation).

Ligand dependent activation of ER was seen robustly at Class A sites, but displayed no evidence of cyclical binding. We propose instead that ER activation occurs rapidly, within 10 min and binding shows no significant change after this point. The two examples we demonstrated — of increasing or decreasing ER binding after activation at the *SNX24* and *ACKR3* TSS (*Figure 4A*) — should be interpreted with caution as, while downstream effects are likely to modulate ER binding, searching for individual outliers results within a large data set will generate false positives. Nonetheless, the two examples imply a secondary level of modulation does occur as previously seen, but at much lower magnitude than proposed in studies focused on ER cycling.

It is possible that alternative conditions may be able to induce tightly regulated cycling; however, we feel this is unlikely in terms of physiology. For example, the work of Metivier et al. makes use of α-amanitin, a RNA polymerase inhibitor. Within the cancer biology setting, these conditions have no direct interpretation. Worse, the mode of action is downstream of the ER and therefore is a confounding factor, not a clear method of synchronization.

In light of our results and the lack of consistency of published results, we propose that the previously described cyclical response kinetics are likely an artefact of observing a highly variable process without replicates. With replicates, the cyclical effect is lost when averaging. Even if a cyclical response existed, our results indicate that it is not regulated tightly enough to be coherently visible across multiple replicates. The variance in ER binding may better be described by heterogeneity in

the cell populations before induction and by current models regarding expression noise as an indicator for greater transcription responsiveness (*Morgan and Marioni, 2018*). Finally, our proposal provides, for the first time, a model that reconciles ChIP-seq data with the stochastic model of nuclear receptor binding as proposed and visualized by those undertaking single model imaging. In contrast, ER cycling has always been irreconcilable with these alternative forms of data (*Lenstra et al., 2016*)

While we cannot discount that our cells could have specifically lost the ability to regulate ER binding in the manner previously described, we have minimized this possibility through the use of cells direct from ATCC, by confirming the cell line by STR genotype and applying the latest methods (*Ben-David et al., 2018*) to confirm that our cell line is genetically similar to the strains used in other labs. Nonetheless, we would welcome further replication of this study.

In summary, through the use of stringent internal controls, we have reproducibly shown that estra-2-diol responsive ER binding is sustained and not cyclical, with the magnitude of the binding primarily defined by the conservation of the ERE at the binding site.

## Materials and methods

### Cell culture

MCF7 cells (RRID:CVCL_0031) were obtained from ATCC and confirmed by STR genotype before culture. For each immunoprecipitation, cells from $2 \times 15$ cm dishes were used. In each 15 cm plate, $2 \times 10^6$ were seeded and grown for 3 days in DMEM (Glibco) with 10% FBS before washing with phosphate buffered saline. Media was replaced with charcoal stripped and phenol red-free DMEM medium. Media was replaced daily for 4 days to ensure removal of estrogenic compounds. Plates were stimulated on day 5 with a final concentration of 100 nM estra-2-diol in EtOH before crosslinking at the required time. All six replicates were done on different dates and represent different passages.

### Cell lines

MCF7 cells were obtained from ATCC. The cell line was authenticated using STR profiling and are confirmed Mycoplasma free.

### pfChIP-seq

Parallel-factor ChIP-seq was performed as previously described (*Guertin et al., 2018*). CTCF antibody was D31H2 Lot:3 (RRID:AB_2086791, Cell Signaling). ER antibody was 06–965 Lot:3008172 (Millipore).

### Data analysis

Reads were aligned using BWA (*Li and Durbin, 2009*), and ENCODE blacklist regions (*ENCODE Project Consortium et al., 2012*) were removed as previously described (*Carroll et al., 2014*). Duplicate reads were removed and peak calling was undertaken using MACS2 (*Zhang et al., 2008*; *Feng et al., 2012*). ER and CTCF peaks were filtered according to the pfChIP-seq protocol (*Guertin et al., 2018*), before normalization and differential binding analysis with Brundle/DiffBind (*Guertin et al., 2018*; *Ross-Innes et al., 2012*) in R. t-SNE plots were generated with Rtsne (*Krijthe, 2015*). Perplexity was tested from 2 to 200 to confirm the stability of the transformation of the data into 2-dimensional space (*Figure 3—figure supplement 1*). Lower perplexities, 2 and 5, gave minimal structure. For perplexities tested between 30 and 200, two stable trajectories were seen in all cases. GREAT (*Welch et al., 2014*) was used to analyze Class B binding sites. Band intensities from previously published studies were measured with ImageJ (*Schneider et al., 2012*).

### Data repositories

Sequencing data have been deposited in GEO under accession code GSE119057.

## Acknowledgments

Experimental Design, Sample Preparation and Data Analysis, ANH. Sample Preparation, AEC. Manuscript preparation ANH, AEC and FM. We thank the CRUK Genomics Core for undertaking the library preparation and sequencing.

## Additional information

### Funding

| Funder | Grant reference number | Author |
| --- | --- | --- |
| Cancer Research UK | C14303/A17197 | Florian Markowetz |
| Breast Cancer Now | 2012NovPR042 | Florian Markowetz |
| Cancer Research UK | C60571/A24631 | Andrew N Holding |
| Cancer Research UK | A19274 | Florian Markowetz |
| Alan Turing Institute | EPSRC grant EP/N510129/ 129/1 | Andrew N Holding |

The funders had no role in study design, data collection and interpretation, or the decision to submit the work for publication.

### Author contributions

Andrew N Holding, Conceptualization, Resources, Data curation, Software, Formal analysis, Supervision, Funding acquisition, Validation, Investigation, Visualization, Methodology, Writing—original draft, Project administration, Writing—review and editing; Amy E Cullen, Investigation, Reviewed experimental design and undertook key experimentation; Florian Markowetz, Supervision, Funding acquisition, Writing—review and editing

### Author ORCIDs

Andrew N Holding http://orcid.org/0000-0002-8459-7048
Amy E Cullen http://orcid.org/0000-0002-5015-1355
Florian Markowetz http://orcid.org/0000-0002-2784-5308

### Decision letter and Author response

Decision letter https://doi.org/10.7554/eLife.40854.028
Author response https://doi.org/10.7554/eLife.40854.029

## Additional files

### Supplementary files

• Source code 1. R Markdown source code for the analysis of the *Supplementary files 1–6*.
DOI: https://doi.org/10.7554/eLife.40854.015

• Supplementary file 1. Table of ER pfChIP-Seq binding matrix.
DOI: https://doi.org/10.7554/eLife.40854.016

• Supplementary file 2. Table of GREAT analysis results.
DOI: https://doi.org/10.7554/eLife.40854.017

• Supplementary file 3. Table of CTCF pfChIP-Seq binding matrix.
DOI: https://doi.org/10.7554/eLife.40854.018

• Supplementary file 4. Quantitative analysis of published ER time series data.
DOI: https://doi.org/10.7554/eLife.40854.019

• Supplementary file 5. Table of ER ChIP-Seq binding matrix without parallel-factor normalization.
DOI: https://doi.org/10.7554/eLife.40854.020

• Supplementary file 6. Table of CTCF ChIP-Seq binding matrix without parallel-factor normalization.
DOI: https://doi.org/10.7554/eLife.40854.021

• Transparent reporting form
DOI: https://doi.org/10.7554/eLife.40854.022

### Data availability

Sequencing data have been deposited in GEO under accession code GSE119057.

The following dataset was generated:

| Author(s) | Year | Dataset title | Dataset URL | Database and Identifier |
|---|---|---|---|---|
| Holding AN, Cullen AE, Markowetz F | 2018 | Genome-wide Estrogen Receptor-alpha activation time-course | https://www.ncbi.nlm.nih.gov/geo/query/acc.cgi?acc=GSE119057 | NCBI Gene Expression Omnibus, GSE119057 |

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
