## [Decision Letter]

Thank you for sending your article entitled "Genome-wide Estrogen Receptor-α activation is sustained, not cyclical" for peer review at *eLife*. Your article is being evaluated by Jessica Tyler as the Senior Editor, a Reviewing Editor, and three reviewers.

The existence of oestrogen receptor cycling has become a debated matter in recent years. Previously it was shown that ERa and its cofactors bind to response elements in a co-ordinated cyclical fashion, but recent data has questioned these findings. In this study, Holding et al., have performed a high number of replicates and included a number of controls as well as a new normalization protocol to demonstrate that ERa binds to DNA rapidly and remains bound, rather than cycling on and off of DNA. If substantiated, this would be an important addition to the understanding of ER biology and regulation. However, substantial criticisms have been raised by several referees around whether cycling indeed does not occur that would need to be addressed in detail.

Essential revisions:

1) The authors normalised the ER ChIP-seq data over CTCF ChIP-seq data, aiming to minimise noise in the analyses. The authors claim that other papers that have described a cyclic behaviour of ER are describing an artefact due to low sample numbers and noisy data. However, the current dataset is quite heavily pre-processed, rendering it impossible to assess whether the hypothesis is correct (cyclic ER pattern, due to noise). The authors should visualise and use the individual datastreams (ER ChIP-seq, CTCF ChIP-seq) and determine whether the ER ChIP-seq data alone would have the cyclic behaviour, or that individual replicates may have that. I find this a key issue.

2) The paper is quite strongly TFF1-focussed, while full genome ER ChIP-seq data is available. With that, I believe the study should be re-positioned, making use of the actual genome-wide information instead.

3) Figure 1: The authors conclude that the data does "not demonstrate evidence of oscillatory binding…" while noting, "there is significantly more variance (F-test, time points >= 10 minutes, p-value < 1 × 10−10) in the ER binding data than in CTCF binding between replicates." However, even allowing for this, as seen in Figure 1, there does appear to be cyclical, dynamic changes in ER binding, particularly when compared to the profile seen for the CTCF control, which appears to be relatively constant. The pattern suggests low ER binding at 20,40 and 60 minutes, and higher binding at 30, 50 and 70 minutes. While not pronounced, this pattern appears to have been lost following analysis. Related to this, is CTCF a good control in this setup? ER and CTCF have been described to cooperate (Ross-Innes et al., 2011). With a biological and functional connection between both proteins, is it really justified to use CTCF to normalise ER data?

4) Over time, some replicates are substantially more variable at specific time points as compared to others, e.g. Figure 1—figure supplement 4 (20, 40, 70, 90 minutes highly variable, while 10, 30, 80 minutes have hardly any variation). Wouldn't this actually support a cyclic patterns in the data? Why would specific time points have a larger variation than others?

5) Figure 2: In addition to the general increase in binding across all sites in response to E2, Part A shows blocks of distinct high affinity ER binding sites that change during the time course. These blocks can still be seen when normalized for CTCF binding, as seen in Part B. The significance of these blocks, particularly in relation to their appearance during the time-course are not adequately discussed.

Figure 2: In addition to the general increase in binding across all sites in response to E2, Part A shows examples of binding sites which appear to exhibit cyclical binding ER in the time-course. These can still be seen when normalized for CTCF binding, as seen in Part B, although, these are difficult to see here, as the normalization appears to dampen the dynamic range for ER binding. The significance of these sites particularly as regards the overall conclusions of the paper are not adequately discussed.

6) I am concerned with the analyses carried in subsection "Quantitative re analysis of independent studies", some of which relies on image analysis of published figures. Unless these analyses were carried out on the primary image data, I would not expect this to be a reliable approach. Also, as the authors themselves point out, there are notable differences in the way the time-course experiments have been carried out in the current study, when compared to the older studies considered in this section. Related to this, In the work by Metivier et al., (2003), the group used α-amanitin to synchronise the cells and strip ERa from response elements. This harsher synchronisation of the cells may result in a clearer cycling of ERa. To be able to compare the data presented here to this previous work, I feel that it will be necessary to perform a ChIP time course for the TFF1 promoter +/- α-amanitin to see if this explains the difference in results.

*Reviewer #1:*

In the field of Estrogen Receptor genomics, an often-reported phenomenon is that ER is cycling on the genome upon E2 stimulation. The authors now claim this is not the case, and the original observations may be related to lower number of replicates and noise in the analyses. Even though this is a compelling concept, I am personally not convinced this is the case, based on the data that are being presented in this manuscript.

1) The authors normalised the ER ChIP-seq data over CTCF ChIP-seq data, aiming to minimise noise in the analyses. The authors claim that other papers that have described a cyclic behaviour of ER are describing an artefact due to low sample numbers and noisy data. However, the current dataset is quite heavily pre-processed, rendering it impossible to assess whether the hypothesis is correct (cyclic ER pattern, due to noise). The authors should visualise and use the individual datastreams (ER ChIP-seq, CTCF ChIP-seq) and determine whether the ER ChIP-seq data alone would have the cyclic behaviour, or that individual replicates may have that. I find this a key issue.

2) The paper is quite strongly TFF1-focussed, while full genome ER ChIP-seq data is available. With that, I believe the study should be re-positioned, making use of the actual genome-wide information instead.

3) Is CTCF a good control in this setup? ER and CTCF have been described to cooperate (Ross-Innes et al., 2011). With a biological and functional connection between both proteins, is it really justified to use CTCF to normalise ER data?

4) Over time, some replicates are substantially more variable at specific time points as compared to others, e.g. Figure 1—figure supplement 4 (20, 40, 70, 90 minutes highly variable, while 10, 30, 80 minutes have hardly any variation). Wouldn't this actually support a cyclic pattern in the data? Why would specific time points have a larger variation than others?

*Reviewer #2:*

The existence of oestrogen receptor cycling has become a debated matter in recent years. Previously it was shown that ERa and its cofactors bind to response elements in a co-ordinated cyclical fashion, but recent data has questioned these findings. In this study, Holding et al., have performed a high number of replicates and included a number of controls to demonstrate that ERa binds to DNA rapidly and remains bound, rather than cycling on and off of DNA. This is important because it furthers our understanding of how nuclear receptors, such as ERa, regulate target gene expression.

The paper is generally sound, but I have one main concern. In the work by Metivier et al., (2003), the group used α-amanitin to synchronise the cells and strip ERa from response elements. This harsher synchronisation of the cells may result in a clearer cycling of ERa. To be able to compare the data presented here to this previous work, I feel that it will be necessary to perform a ChIP time course for the TFF1 promoter +/- α-amanitin to see if this explains the difference in results.

*Reviewer #3:*

The interaction of transcripiton factors with DNA and chromatin are considered to be highly dynamic. This conclusion has been supported by numerous studies, including studies examining nuclear receptor dynamics, such as those previously carried out for Estrogen Receptor α (ER). Both direct evaluation of ER binding by Chromatin Immunopreciptation (ChIP) methods and imaging in live cells have supported a paradigm of rapid and continuous exchange events with the DNA. These highly dynamic interactions are a property of both DNA-protein and protein- protein interactions and are inherent to the transcriptional response.

The paper by Holden et al. seeks to use ChIPseq analysis for the estrogen receptor, in the MCF7 estrogen responsive breast cancer cell line, in order to further investigate ER transcriptional dynamics. Their analysis has used a parallel factor ChIP (pfChIP) normalization methodology, recently published by the lead author. Using this analysis, the authors observe that, rather than cycling at binding sites, a sustained increase in binding (affinity) is seen, together with a class of estrogen (E2) independent binding sites.

The study describes a set of ChIPseq time course data for estrogen receptor, which defines a potentially useful resource. The analysis using pfChIP normalization is interesting. Comments are as follows;

1) Figure 1: The authors conclude that the data does "not demonstrate evidence of oscillatory binding…" while noting, "there is significantly more variance (F-test, time points >= 10 minutes, p-value < 1 × 10−10) in the ER binding data than in CTCF binding between replicates." However, even allowing for this, as seen in Figure 1, there does appear to be cyclical, dynamic changes in ER binding, particularly when compared to the profile seen for the CTCF control, which appears to be relatively constant. The pattern suggests low ER binding at 20,40 and 60 minutes, and higher binding at 30, 50 and 70 minutes. While not pronounced, this pattern appears to have been lost following analysis.

2) Figure 2: In addition to the general increase in binding across all sites in response to E2, Part A shows blocks of distinct high affinity ER binding sites that change during the time course. These blocks can still be seen when normalized for CTCF binding, as seen in Part B. The significance of these blocks, particularly in relation to their appearance during the time-course are not adequately discussed.

3) Figure 2: In addition to the general increase in binding across all sites in response to E2, Part A shows examples of binding sites which appear to exhibit cyclical binding ER in the time-course. These can still be seen when normalized for CTCF binding, as seen in Part B, although, these are difficult to see here, as the normalization appears to dampen the dynamic range for ER binding. The significance of these sites particularly as regards the overall conclusions of the paper are not adequately discussed.

4) I am concerned with the analyses carried in subsection "Quantitative re analysis of independent studies", some of which relies on image analysis of published figures. Unless these analyses were carried out on the primary image data, I would not expect this to be a reliable approach. Also, as the authors themselves point out, there are notable differences in the way the time-course experiments have been carried out in the current study, when compared to the older studies considered in this section.

---

## [Author Response]

Essential revisions:1) The authors normalised the ER ChIP-seq data over CTCF ChIP-seq data, aiming to minimise noise in the analyses. The authors claim that other papers that have described a cyclic behaviour of ER are describing an artefact due to low sample numbers and noisy data. However, the current dataset is quite heavily pre-processed, rendering it impossible to assess whether the hypothesis is correct (cyclic ER pattern, due to noise). The authors should visualise and use the individual datastreams (ER ChIP-seq, CTCF ChIP-seq) and determine whether the ER ChIP-seq data alone would have the cyclic behaviour, or that individual replicates may have that. I find this a key issue.

We thank you for highlighting this point. While we feel the pfChIP-seq is the most reliable method for analysing the data as it controls for IP efficiency, nonetheless presenting it alongside a more classic analysis clearly strengthens the manuscript. We have re-analysed the data using only the ER ChIP-seq data stream and found no significant differences in the results. These data are now provided in Supplementary file 5 and Supplementary file 6 and the analysis is in the Rmarkdown. Before interpreting, we would draw the attention of the reviewers to our analysis in responses 3 and 4 which discuss the significance of the variation of the mean in both plots.

Our analysis of the ER data stream in the same manner as the complete pfChIP-seq data demonstrates little difference between the two and confirms that pfChIP-seq is not masking that possibility of a regulated oscillation in ER binding. Rather, it ensures that the variance is not caused by changes in IP efficiency.

As a result, we are confident that there is no consistent cycling in the individual replicates or at individual sites (illustrated in Panel C for TFF1, and genome wide in panel D).

We have added the following to the text to highlight this result and the figure below as Figure 3—figure supplement 2.

“Undertaking the same analysis of the ER only ChIP-seq data stream gave very similar results to that of pfChIP-seq analysis, confirming that any potential cycling is not suppressed by the method (Figure 3—figure supplement 2). As with the pfChIP-seq analysis, no clear cycling was seen for the individual replicates (Figure 3—figure supplement 2A).”

2) The paper is quite strongly TFF1-focussed, while full genome ER ChIP-seq data is available. With that, I believe the study should be re-positioned, making use of the actual genome-wide information instead.

We introduce the paper with TFF1 as it is the classical ER target gene and the most studied within the literature. As a result, it forms the basis of Figure 1 alongside a supplemental figure for the RARA loci. We do not feel that this makes the paper ‘strongly focused on TFF1’. To the contrary, three-quarters of the manuscript figures are already focused on genome-wide analysis and only 5% of the paper’s text (~180 words) forms the section “ER Binding at the TFF1 Promoter”. Outside this section, the word ‘TFF1’ is only mentioned 6 times. This includes a mention in the abstract, essential given the gene’s importance to the field, and in the section “Quantitative re-analysis of independent studies”.

Figure 2 is genome-wide and compared to public genome-wide data. Figure 3 and Figure 4 present genome-wide information and the final discussion is also focused on the genome-wide results (and does not mention TFF1 at all).

In meeting the other request of reviewers, we have specifically been asked to further analyse the TFF1 promoter data, e.g. point 3 below, and have therefore had to include more on this topic. Even with these additional analyses, the paper is strongly focused on genome-wide results and not on individual loci.

3a) Figure 1: The authors conclude that the data does "not demonstrate evidence of oscillatory binding…" while noting, "there is significantly more variance (F-test, time points >= 10 minutes, p-value < 1 × 10−10) in the ER binding data than in CTCF binding between replicates." However, even allowing for this, as seen in Figure 1, there does appear to be cyclical, dynamic changes in ER binding, particularly when compared to the profile seen for the CTCF control, which appears to be relatively constant. The pattern suggests low ER binding at 20,40 and 60 minutes, and higher binding at 30, 50 and 70 minutes. While not pronounced, this pattern appears to have been lost following analysis.

Residual variability in the estimate of the mean is the expected result for a n=6 experiment with this level of variance. Patterns visible by eye are inevitable and need to be rigorously tested. We find that none of the apparent patterns are statistically significant, as shown by the plot below, with the exception of the comparison between 0 minutes and other time points, which is significant at p < 0.05, and represents the sustained activation of the ER.

The following comment has been added to Figure 1:

“Pairwise comparison found no significant changes in binding at the TFF1 promoter (t-test, two-sided, FDR < 0.05) except for when comparing against the 0 minute time point.”

There is the possibility of a weaker modulation of the signal that we do not have the statistical power to see; however, the timings the reviewer proposes would suggest a completely novel behaviour that has not been described before. There is no statistical evidence for the reviewer’s claim nor does it fit the classical description of cyclical ER binding which we are disputing (maximal binding at 45 minutes, no binding at 90 minutes). We believe our proposal – that the apparent binding dynamics are a result of noise – is superior, as it integrates all previous work and provides a model that fits with the literature.

b) Related to this, is CTCF a good control in this setup? ER and CTCF have been described to cooperate (Ross-Innes et al., 2011). With a biological and functional connection between both proteins, is it really justified to use CTCF to normalise ER data?

Thank you for raising this key point. A good choice of the control factor is critical and should be justified.

The conclusion from Ross-Innes et al., 2011 is as follows: “We now map CTCF binding genome wide in breast cancer cells and find that CTCF binding is unchanged in response to estrogen or tamoxifen treatment.” Further, Ross-Innes et al., state that the overlap of ER and CTCF sites is small. In addition, as described in Guertin et al., 2018, this small number of sites has been excluded from the normalisation process to ensure that no interference occurs. We have added the following to the text to clarify this important point:

“Peak count data from CTCF binding sites were used to normalize between conditions as these sites have previously been shown to be unchanged in response to estrogen (Ross-Innes et al., 2011)”

4) Over time, some replicates are substantially more variable at specific time points as compared to others, e.g. Figure 1—figure supplement 4 (20, 40, 70, 90 minutes highly variable, while 10, 30, 80 minutes have hardly any variation). Wouldn't this actually support a cyclic pattern in the data? Why would specific time points have a larger variation than others?

This experiment was undertaken with 6 replicates, i.e. n=6. While this is much higher than almost all ChIP-seq experiments in the literature, it is still not a large value of n in terms of statistics. When estimating the mean and variance of the population, which is what we are actual plotting in a box plot, the accuracy of estimating the variance is more susceptible to low n than the mean. Therefore, we would expect each time point to have a different estimate for the variance if the data is noisy and n is relatively small. This is true even if the means are similar.

We believe the challenge is that the eye is very much drawn to finding patterns, even if those patterns occur by chance, which is exactly what happens when sampling 6 data points from an underlying distribution. Therefore, to robustly address your concerns, we undertook an F-test between every time point (Figure 1). We found no significant difference in variance (heatmap below, left), implying that the changes in variance can be explained by the sampling process of a n=6 experiment.

We have added the following statement:

“In contrast, pairwise F-test (two-sided, FDR < 0.05) for ER binding at all time points showed no significant difference in the variance for any comparison.”

**Author response image 2. respfig2:** 

5a) Figure 2: In addition to the general increase in binding across all sites in response to E2, Part A shows blocks of distinct high affinity ER binding sites that change during the time course. These blocks can still be seen when normalized for CTCF binding, as seen in Part B. The significance of these blocks, particularly in relation to their appearance during the time-course are not adequately discussed.

We have now added the following to further discuss this result:

“While grouping by maximum occupancy in Figure 2A was essential to highlight these features in the context of Guertin et al.*,’* s previous study, in our analysis we found this method distorts the data and the effects that drove the appearance of blocks are not statistically significant (Figure 2—figure supplement 1) with the exception of the 0 to 10 minute contrast.”

b) Figure 2: In addition to the general increase in binding across all sites in response to E2, Part A shows examples of binding sites which appear to exhibit cyclical binding ER in the time-course. These can still be seen when normalized for CTCF binding, as seen in Part B, although, these are difficult to see here, as the normalization appears to dampen the dynamic range for ER binding. The significance of these sites particularly as regards the overall conclusions of the paper are not adequately discussed.

We thank the reviewers for highlighting these peaks. However, it should be noted that Figure 2A is much more aggressively normalised and results in considerable distortion of the data compared to Part B, not less.

We therefore have added the following to the manuscript.

“As far as we are aware, the loss on ER binding on activation with estra-2-diol is unprecedented, and therefore the presence of a block of ER sites with maximal binding at 0 minutes warranted deeper investigation. Analysis of the class average (Figure 2—figure supplement 2A) showed that the variance of this class is much greater than the decrease seen between 0 and 10 minutes. A more detailed analysis of the individual trajectories of each binding site (Figure 2—figure supplement 2B) showed a similar result, with the maximal binding at time zero appearing marginal and in all cases. We therefore concluded that maintaining the group of the blocks between Figure 2A and B visually overemphasized this feature within the data.”

6a) I am concerned with the analyses carried in the section "Quantitative re analysis of independent studies", some of which relies on image analysis of published figures. Unless these analyses were carried out on the primary image data, I would not expect this to be a reliable approach.

The reviewer raises a good point about the quality of the data. Two of the studies used qPCR and the numerical values could be obtained from graphs within the papers. This information is supplied in the Supplementary Files. The remaining two studies did require image analysis, which has challenges. Nonetheless, this will not change the results for three key reasons:

A) The aim of the original figure was to accurately represent the data, and this is the only form it was reported in. Therefore, information can be gained from analysing these images.

B) The images do not to appear to be significantly altered for contrast. While contrast editing is possible, our argument is based on the *relative* magnitude between time points, which will not change on increasing contrast.

C) The most likely effects of image analysis is a loss of dynamic range or oversaturation. These effects would broaden the peak of any cycling event and would not affect our conclusion. We see no correlation in the timing of binding events at all.

Our analysis makes the best of previously published results, rather than disregarding them as useless. By using a quantitative analysis of the published image, we have taken an unbiased approach and darker bands clearly provide more signal when our graph is compared to the original studies. As none of the events previously reported correlate, even on a qualitative level, we believe this is an important part of the study and we have been completely honest on our methods. To reiterate, the alternative is to discount all previous work, which we believe would find less support.

We have added the following statement to make the limitations clear:

“Quantitative image analysis presents limitations in re-establishing the exact values without the primary data; however, the primary data was not available. In lieu of this, analysis of the published data was considered adequate as the relative intensities will be preserved.”

b) Also, as the authors themselves point out, there are notable differences in the way the time-course experiments have been carried out in the current study, when compared to the older studies considered in this section.

We apologise that the manuscript was not clear. It is for exactly these reasons that we identified datasets that used identical conditions. This enabled us to ensure these studies were comparable to our own study. We have clarified this in the text as follows:

“We identified several studies that undertook the same analysis in the MCF7 cell line, with the same concentration of estra-2-diol, same crosslinking time scale, and at the same promoter as the best datasets for comparison with each other and to our dataset.”

c) Related to this, In the work by Metivier et al., (2003), the group used α-amanitin to synchronise the cells and strip ERa from response elements. This harsher synchronisation of the cells may result in a clearer cycling of ERa. To be able to compare the data presented here to this previous work, I feel that it will be necessary to perform a ChIP time course for the TFF1 promoter +/- α-amanitin to see if this explains the difference in results.

Exactly for the reasons the reviewer expressed in point 6b, we chose not to use α-amanitin.

Few studies beyond those of Metivier and his co-workers have made use of these conditions. From a cancer biology point of view, the effects of α-amanitin, which is deadly to organisms (LD 50 0.1mg/kg), has no physiological interpretation. Further, α-amanitin blocks the polymerase which is downstream of the estrogen receptor. There is no reason to believe that it directly impacts the ER and our study does not look at RNA synthesis.

Additionally, α-amanitin does not provide a mechanism to synchronise cell cycle progression. It will only at best halt the cell in the phase it is in. Estrogen starvation for multiple days has the same result, causing MCF7 (and other ER+ cancer models) cells to stop cell division. Synchronisation has clearly occurred as the 0-10 minute response to E2 is conversed genome-wide and between replicates.

One solution to these challenges is single molecule imaging strategies, as used by the Gordon Hager Lab. Gordon Hager has long maintained that the ER cycling model is not supported by his work and is clearly incompatible with the results he has presented. This contradiction is described by Lenstra et al., (2016). In contrast, our model is compatible with the work of the Hager Lab.

Therefore, while we accept that our study does not rule out that α-amanitin may cause the effects seen in Metiever’s manuscript, we can rule out that this does not occur in more physiological conclusions and that our results best fit with the wider literature.

We have added the following paragraph to accept that we cannot rule out that α-amanitin would change the results:

“It is possible that alternative conditions may be able to induce tightly regulated cycling; however, we feel this is unlikely in terms of physiology. For example, the work of Metivier et al., makes use of α-amanitin, an RNA polymerase inhibitor. Within the cancer biology setting, these conditions have no direct interpretation. Worse, the mode of action is downstream of the ER and therefore is a confounding factor, not a clear method of synchronisation.”

And later:

“Finally, our proposal provides, for the time, a first model that reconciles ChIP-seq data with the stochastic model of nuclear receptor binding as proposed and visualized by those undertaking single model imaging. In contrast, ER cycling has always been irreconcilable with these alternative forms of data (Lenstra et al., 2016).”